# LEARNING WHEN TO BE SPARSE: ADAPTIVE ACTIVATIONS VIA TWO-PARAMETER ENTROPY

**Roman Rudamenko**
Lomonosov Moscow State University

**Dmitry Abulkhanov**
MBZUAI

**Konstantin Semenov**
Lomonosov Moscow State University

**Michael Diskin**
HSE University

**Alexander Savchenko**
Lomonosov Moscow State University

## ABSTRACT

The softmax operator, while foundational to modern machine learning, arises from Shannon entropy regularizationan assumption that breaks down for systems with long-range correlations or power-law tails. Sparse alternatives such as $\alpha$-entmax address this via Tsallis entropy, but rigidly tie sparsity to a single parameter. We introduce SHARMIX, a two-parameter activation based on Sharma–Mittal entropy that unifies the Shannon, Rényi, and Tsallis families. We derive closed-form, Lipschitz-continuous Jacobians for both input logits and entropy parameters $(q, r)$, enabling end-to-end learning via implicit differentiation. This allows SHARMIX to dynamically adapt to the statistical properties of the data. Experiments on text classification, CIFAR-100, and ImageNet-1k demonstrate that SHARMIX automatically navigates the accuracy-sparsity trade-off.

## 1 INTRODUCTION

The softmax operator emerges as the unique solution to the maximum Shannon entropy problem

$$\mathbf{p}^\star = \arg\max_{\mathbf{p} \in \Delta^N} \mathbf{p}^\top \mathbf{z} + H_{\text{Sh}}(\mathbf{p}), \tag{1}$$

where $\Delta^N$ is the probability simplex and $H_{\text{Sh}}(\mathbf{p}) = -\sum_i p_i \log p_i$ is Shannon entropy.

However, softmax's strict positivity yields dense distributions that can over-allocate mass to improbable classes, particularly in heavy-tailed regimes where Shannon entropy fails to capture power-law statistics Tsallis (1988). The $\alpha$-entmax family Blondel et al. (2019), rooted in Tsallis entropy, produces sparse outputs but the single parameter $\alpha$ rigidly couples sparsity to curvature: values appropriate for heavy-tailed data prove excessive for balanced distributions.

To overcome this limitation, we introduce **SharMiX**, a differentiable mapping based on Sharma–Mittal entropy $H_{q,r}$, which generalizes Tsallis ($r \to q$) and Rényi ($r \to 1$) as limits. SharMiX solves

$$\mathbf{p}^\star = \arg\max_{\mathbf{p} \in \Delta^N} \mathbf{p}^\top \mathbf{z} + \gamma H_{q,r}(\mathbf{p}). \tag{2}$$

By jointly optimizing $q$ and $r$ during training, SharMiX adaptively interpolates between sparse and dense regimes.

**Contributions:**

  (i) We propose SHARMIX, a differentiable probability mapping derived from Sharma–Mittal entropy that unifies softmax, $\alpha$-entmax, and Rényi-type activations (Section 4).

 (ii) We derive closed-form Jacobians with respect to both logits and entropy parameters $(q, r)$, enabling end-to-end learning (Section 4.2).

(iii) We demonstrate empirically that SharMiX adapts to the data distribution across different statistical regimes (Section 5).

## 2  RELATED WORK

Temperature scaling $\mathrm{softmax}(\mathbf{z}/T)$ provides calibration (Hinton et al., 2015; Guo et al., 2017), but remains fundamentally dense. Sparsemax (Martins & Astudillo, 2016) introduced exact zeros via Tsallis entropy, and $\alpha$-entmax (Blondel et al., 2019) generalized this to arbitrary $\alpha > 1$. Subsequent work learns $\alpha$ per head (Correia et al., 2019). The Fenchel–Young loss framework (Blondel et al., 2020) unified entropy-regularized prediction with convexity guarantees. The Sharma–Mittal entropy (Sharma & Mittal, 1975) provides a two-parameter generalization, yet no prior work has constructed a practical neural network layer from it with efficient gradients for both parameters.

## 3  BACKGROUND

### 3.1  GENERALIZED ENTROPIES

Classical Shannon entropy assumes *extensivity*: $H(A \cup B) = H(A) + H(B)$ for independent systems. This fails for heavy-tailed distributions with tail index $\zeta$ where higher moments diverge. The Sharma–Mittal entropy addresses this:

$$H_{q,r}(\mathbf{p}) = \frac{1}{1-r}\left[\left(\sum_i p_i^q\right)^{\frac{1-r}{1-q}} - 1\right], \quad q > 0,\ r \neq 1. \tag{3}$$

This unifies Rényi entropy ($r \to 1$) and Tsallis entropy ($r \to q$) as special cases (see Appendix A for derivations).

**Theorem 3.1** (Non-Extensivity in Heavy-Tailed Systems). *For i.i.d. samples from a Pareto distribution with tail index $\zeta \in (1, 2]$, the empirical q-mean converges almost surely for $q < \zeta$, whereas the classical law of large numbers fails. Systems with $\zeta \leq 2$ are non-extensive, and $H_{q,r}$ with $q > 1$ stabilizes optimization by down-weighting frequent modes.*

The proof appears in Appendix B.2.

### 3.2  FROM SOFTMAX TO SPARSE ACTIVATIONS

Replacing Shannon entropy with Tsallis entropy $H_q^{\mathrm{Ts}}(\mathbf{p}) = \frac{1 - \sum_i p_i^q}{q-1}$ in the MaxEnt problem yields the $\alpha$-entmax solution:

$$p_i^\star \propto \left[(z_i - \tau)_+\right]^{\frac{1}{q-1}}, \tag{4}$$

where $\tau$ enforces normalization. Crucially, Tsallis and Rényi entropies produce *equivalent* solutions up to reparameterization (Lemma 4.1), meaning one-parameter entropies cannot decouple sparsity from curvature.

## 4  METHOD

### 4.1  SHARMIX ACTIVATION

The SharMiX solution has the thresholded power-law form:

$$p_i^\star = \frac{\left[\alpha\,(z_i - \tau)\right]_+^\beta}{\sum_{j=1}^N \left[\alpha\,(z_j - \tau)\right]_+^\beta}, \quad \beta = \frac{1}{q-1}, \tag{5}$$

where $\tau$ enforces normalization. The parameter $\beta$ controls sparsity: $\beta = 1$ yields sparsemax, $\beta \to \infty$ yields softmax, and $\beta \to 0^+$ yields hard argmax (Proposition **??** in Appendix B.3).

**Lemma 4.1** (Equivalence of Tsallis and Rényi). *For any $q > 1$, on regions with fixed active set, the maximizers under Tsallis or Rényi entropy induce the same power-law family up to rescaling of $\gamma$. Their input Jacobians take the same functional form.*

See Appendix B.4 for the proof.

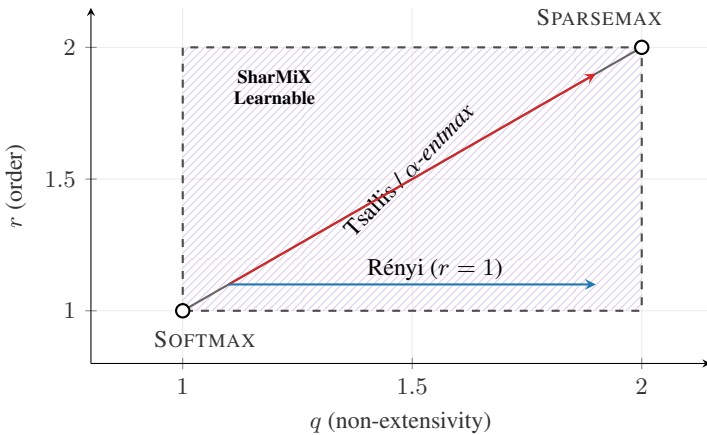

Figure 1: The Sharma-Mittal parameter space. SharMiX recovers Tsallis/$\alpha$-entmax along $r = q$, Rényi along $r = 1$, and softmax at $(1, 1)$.

## 4.2 JOINT $(q, r)$ OPTIMIZATION

To learn $(q, r)$ end-to-end, we differentiate the normalization constraint implicitly. Define $f(\tau, q, r; \mathbf{z}) = \sum_i p_i^\star - 1 = 0$. By the implicit function theorem, $\tau = \iota(q, r; \mathbf{z})$ locally.

The key insight is expressing $p_i^\star$ as softmax over transformed logits $F_i = \frac{\log[(q-1)(z_i-\tau)]_+}{q-1}$, enabling direct application of softmax Jacobian structure. For any loss $\mathcal{J}(\mathbf{p}^\star)$:

$$\frac{\partial \mathcal{J}}{\partial q} = \nabla_{\mathbf{p}^\star} \mathcal{J}^\top \left( \frac{\partial \mathbf{p}^\star}{\partial q} + \frac{\partial \mathbf{p}^\star}{\partial \tau} \frac{\partial \tau}{\partial q} \right), \tag{6}$$

$$\frac{\partial \mathcal{J}}{\partial r} = \nabla_{\mathbf{p}^\star} \mathcal{J}^\top \left( \frac{\partial \mathbf{p}^\star}{\partial r} + \frac{\partial \mathbf{p}^\star}{\partial \tau} \frac{\partial \tau}{\partial r} \right). \tag{7}$$

The complete closed-form expressions appear in Appendix C.

**Theorem 4.2** (Differentiability and Computational Guarantees). *For $q, r \in [1 + \delta, 2]$ ($\delta > 0$), the mapping $(\mathbf{z}, q, r) \mapsto \mathbf{p}^\star$ is $C^1$ with Lipschitz Jacobians $\|\partial \mathbf{p}^\star / \partial \theta\| \leq LN$. Forward/backward passes are $\mathcal{O}(N \log \epsilon^{-1})$ and $\mathcal{O}(N)$, respectively.*

See Appendix B.5 for the proof.

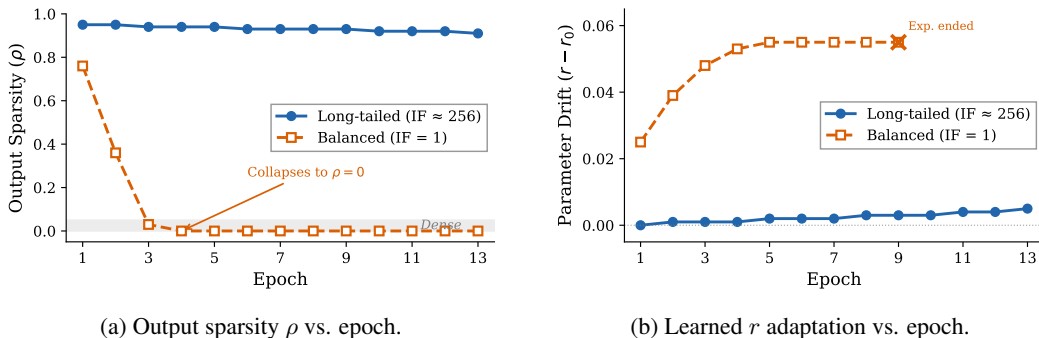

(a) Output sparsity $\rho$ vs. epoch.  (b) Learned $r$ adaptation vs. epoch.

Figure 2: **SharMiX adaptation on ImageNet-1k.** Long-tailed regime maintains high sparsity ($\rho > 0.91$); balanced regime collapses to dense ($\rho \to 0$) by epoch 4.

Table 1: Test metrics on Airline Tweets. Mean $\pm$ std over five runs.

| Method | Accuracy | Macro-$F_1$ | Sparsity |
|---|---|---|---|
| **SharMiX** $(1.25, 1.50)$ | $\mathbf{0.910 \pm 0.006}$ | $\mathbf{0.880 \pm 0.009}$ | 0.000 |
| SharMiX $(1.50, 1.50)$ | $0.907 \pm 0.007$ | $0.876 \pm 0.008$ | 0.000 |
| Softmax | $0.893 \pm 0.007$ | $0.855 \pm 0.010$ | 0.000 |
| Sparsemax | $0.908 \pm 0.004$ | $0.876 \pm 0.007$ | 0.389 |
| Entmax$_{1.50}$ | $0.600 \pm 0.009$ | $0.463 \pm 0.014$ | 0.631 |

Table 2: CIFAR-100 final metrics (ResNet-18, epoch 50).

| Method | Val. Acc. | Sparsity |
|---|---|---|
| SharMiX fixed $(1.25, 1.5)$ | **0.484** | 0.000 |
| Softmax | 0.474 | 0.000 |
| SharMiX learned | 0.428 | 0.005 |
| Sparsemax | 0.404 | 0.860 |

## 5 EXPERIMENTS

We evaluate SharMiX on three benchmarks: Airline Tweet Sentiment (imbalanced, $\zeta \approx 1.8$), CIFAR-100 (balanced), and ImageNet-1k (both regimes). Implementation details appear in Appendix E.

### 5.1 RESULTS

**Imbalanced Text (Table 1).** SharMiX achieved the best performance (0.910 accuracy), outperforming softmax by $+1.7$ pp while converging to dense outputs. Fixed $\alpha$-entmax failed due to excessive sparsity.

**Balanced Images (Table 2).** On CIFAR-100, SharMiX fixed $(1.25, 1.5)$ outperformed softmax by $+1.0$ pp with zero sparsity. Sparsemax underperformed by $-7.0$ pp due to enforced sparsity. Extended results in Appendix D.

**ImageNet-1k (Figure 2).** On the long-tailed split, sparsity remained high ($\rho = 0.91$–$0.95$). On the balanced split, sparsity collapsed to zero by epoch 5. The parameter $r$ exhibited the primary adaptation, while $q$ remained stable.

## 6 DISCUSSION

Our results highlight three key behaviors. **First**, SharMiX adapts its sparsity to the data distribution: high sparsity on long-tailed ImageNet, dense outputs on balanced CIFAR-100 and Airline Tweets. **Second**, the Sharma–Mittal form improves performance even without sparsitySharMiX outperformed softmax on both datasets despite dense outputs. **Third**, the parameters serve distinct roles: $r$ drives adaptation while $q$ remains stable.

**Limitations.** The benchmarks represent a limited sample of distributions. Learned parameters showed initialization sensitivity (Table 1 in Appendix). The parameter ranges and learning rates were not systematically optimized.

## 7 CONCLUSION

We introduced SharMiX, a two-parameter activation derived from Sharma–Mittal entropy that unifies softmax, sparsemax, and $\alpha$-entmax. By deriving closed-form Jacobians for both $(q, r)$, we

enabled end-to-end learning of sparsity and curvature. Experiments demonstrate automatic adaptation to class-frequency distributions. Future work may explore per-layer parameterization and application to attention mechanisms.

## REFERENCES

Mathieu Blondel, Vlad Niculae, Takuma Ohto, and Naonori Ueda. Learning sparse projections with $\alpha$-entmax. In *International Conference on Learning Representations (ICLR)*, 2019.

Mathieu Blondel, Quentin Berthet, and Vlad Niculae. Fenchel–young losses: A vector space view. In *Advances in Neural Information Processing Systems (NeurIPS)*, 2020.

Guilherme M. Correia, Viorel Niculae, and André F. T. Martins. Adaptively sparse transformers. In *Proceedings of the Conference on Empirical Methods in Natural Language Processing*, 2019.

Chuan Guo, Geoff Pleiss, Yu Sun, and Kilian Q Weinberger. On calibration of modern neural networks. In *International conference on machine learning*, pp. 1321–1330. PMLR, 2017.

Geoffrey Hinton, Oriol Vinyals, and Jeff Dean. Distilling the knowledge in a neural network. *arXiv preprint arXiv:1503.02531*, 2015.

Andre Martins and Ramon Astudillo. From softmax to sparsemax: A sparse model of attention and multi-label classification. In *International conference on machine learning*, pp. 1614–1623. PMLR, 2016.

B. D. Sharma and D. P. Mittal. New nonadditive measures of inaccuracy. *Journal of Mathematical Sciences*, 1975.

C. Tsallis. Possible generalization of boltzmann–gibbs statistics. *Journal of Statistical Physics*, 1988.

## A  ENTROPY LIMIT DERIVATIONS

The Sharma–Mittal entropy contains both Rényi and Tsallis entropies as special limiting cases. Let $S_q(\mathbf{p}) = \sum_i p_i^q$.

**Rényi Entropy** ($r \to 1$): Applying L'Hôpital's rule:

$$\lim_{r \to 1} \frac{S_q^{\frac{1-r}{1-q}} - 1}{1 - r} = \frac{\ln(S_q)}{1 - q} = \frac{1}{1 - q} \log\left(\sum_i p_i^q\right) = H_q^{\mathrm{Rey}}(\mathbf{p}) \tag{8}$$

**Tsallis Entropy** ($r \to q$): As $r \to q$, the exponent $\frac{1-r}{1-q} \to 1$:

$$\lim_{r \to q} H_{q,r}(\mathbf{p}) = \frac{1}{1 - q}\left(\sum_i p_i^q - 1\right) = H_q^{\mathrm{Ts}}(\mathbf{p}) \tag{9}$$

## B  DEFERRED PROOFS

### B.1  PROOF OF PROPOSITION: BREGMAN DIVERGENCE GENERALIZATION

**Proposition B.1** (Bregman Divergence Generalization). *For a strictly convex $H : \Delta^N \to \mathbb{R}$, the entmax solution $\mathbf{p}^\star = \arg\max_{\mathbf{p}} \mathbf{p}^\top \mathbf{z} + H(\mathbf{p})$ is the proximal operator $\mathrm{prox}_{\gamma H^\star}(\mathbf{z})$, where $H^\star(\mathbf{u}) = \sup_{\mathbf{p}} \mathbf{p}^\top \mathbf{u} - H(\mathbf{p})$ is the convex conjugate. The Fenchel–Young loss $\mathcal{L}(\mathbf{z}, \mathbf{y}) = H^\star(\mathbf{z}) + H(\mathbf{y})$ is convex in $\mathbf{z}$.*

*Proof.* By the Fenchel-Moreau theorem, for a closed convex function $H$, we have $H = (H^\star)^\star$. The proximal operator of $H^\star$ is:

$$\mathrm{prox}_{H^\star}(\mathbf{z}) = \arg\min_{\mathbf{u}} \frac{1}{2}\|\mathbf{u} - \mathbf{z}\|^2 + H^\star(\mathbf{u})$$

For entropy regularization on the simplex, the optimality condition for $\mathbf{p}^\star = \arg\max_{\mathbf{p} \in \Delta^N} \mathbf{p}^\top \mathbf{z} + H(\mathbf{p})$ yields $\mathbf{z} + \nabla H(\mathbf{p}^\star) = \tau \mathbf{1}$ where $\tau$ is the Lagrange multiplier for the simplex constraint.

The convex conjugate $H^\star(\mathbf{u}) = \sup_{\mathbf{p} \in \Delta^N} \mathbf{p}^\top \mathbf{u} - H(\mathbf{p})$ is convex since $H$ is closed and convex. The Fenchel-Young loss $\mathcal{L}(\mathbf{z}, \mathbf{y}) = H^\star(\mathbf{z}) + H(\mathbf{y}) - \mathbf{z}^\top \mathbf{y}$ is convex in $\mathbf{z}$ and equals the Bregman divergence $D_H(\mathbf{y}, \mathbf{p}^\star)$ where $\mathbf{p}^\star = \nabla H^\star(\mathbf{z})$. $\square$

### B.2  PROOF OF THEOREM 3.1

*Proof.* For a Pareto distribution with tail index $\zeta$, the density is $f(x) = \zeta x_m^\zeta x^{-\zeta-1}$ for $x \geq x_m$. The $q$-th moment exists iff $q < \zeta$:

$$\mathbb{E}[X^q] = \int_{x_m}^\infty x^q f(x) dx = \frac{\zeta x_m^q}{\zeta - q}, \quad q < \zeta$$

For $\zeta \in (1, 2]$, the mean exists but variance is infinite. The empirical $q$-mean:

$$\hat{\mathbb{E}}_q[X] = \left(\frac{1}{M}\sum_{k=1}^M X_k^q\right)^{1/q} \Bigg/ \left(\frac{1}{M}\sum_{k=1}^M X_k^{q-1}\right)^{1/(q-1)}$$

converges almost surely to $\mathbb{E}_q[X]$ for $q < \zeta$ by the strong law applied to $X^q$ and $X^{q-1}$ (both have finite expectations).

The Sharma-Mittal entropy with $q > 1$ and $r \in (0, 1)$ provides stabilization through $q$-escort probabilities $P_q(x) \propto p(x)^q$, which down-weight the heavy tail, transforming the divergent variance problem into a convergent one in the $q$-deformed space. $\square$

## B.3 PROOF OF PROPOSITION: SPARSITY CONTROL VIA $\beta$

**Proposition B.2** (Sparsity control via $\beta$). *The parameter $\beta = 1/(q-1)$ in the power-law solution controls sparsity:*

*– As $\beta \to 1$ ($\alpha \to 2$): sparsemax, $p_i^\star \propto (z_i - \tau)_+$*
*– As $\beta \to \infty$ ($\alpha \downarrow 1$): $\lim_{\alpha \downarrow 1} p^\star(\alpha) = \text{softmax}(\mathbf{z})$*
*– As $\beta \to 0^+$ ($\alpha \to \infty$): concentration on $\arg\max_i z_i$*

*Proof.* Write $\alpha \equiv q > 1$ and $\beta = 1/(\alpha - 1)$. The solution has form $p_i^\star(\alpha) = \frac{[u_i(\alpha)]_+^\beta}{\sum_j [u_j(\alpha)]_+^\beta}$ with $u_i(\alpha) = (\alpha - 1)(z_i - \tau_\alpha)$.

**Case $\alpha = 2$ ($\beta = 1$).** For $\alpha = 2$: $p_i^\star(2) \propto (z_i - \tau_2)_+$, the sparsemax mapping—the Euclidean projection of $\mathbf{z}$ onto the probability simplex.

**Limit $\alpha \downarrow 1$ ($\beta \to \infty$).** Reparameterize as $F_i(\alpha) = \frac{\log[(\alpha - 1)(z_i - \tau_\alpha)]}{\alpha - 1}$ so $p^\star(\alpha) = \text{softmax}(\mathbf{F}(\alpha))$. As $\alpha \downarrow 1$, the normalization constraint forces $(\alpha - 1)(z_i - \tau_\alpha) = \exp[(\alpha - 1)(z_i - c_\alpha)]$ for some $c_\alpha$, whence $F_i(\alpha) = z_i - c_\alpha$ and therefore $\lim_{\alpha \downarrow 1} p^\star(\alpha) = \text{softmax}(\mathbf{z})$.

**Limit $\alpha \to \infty$ ($\beta \to 0^+$).** Let $z_{\max} = \max_j z_j$ and $\mathcal{A} = \{i : z_i = z_{\max}\}$. For any $i \notin \mathcal{A}$, there exists $\eta_i > 0$ such that $z_i \leq z_{\max} - \eta_i$. Hence $\frac{[u_i(\alpha)]_+^\beta}{[u_{\max}(\alpha)]_+^\beta} \leq \left(1 - \frac{(\alpha - 1)\bar{\eta}}{u_{\max}(\alpha)}\right)^\beta \xrightarrow{\alpha \to \infty} 0$, so mass assigned to $i \notin \mathcal{A}$ vanishes. If $|\mathcal{A}| = 1$, this yields the one-hot vector; if $|\mathcal{A}| > 1$, symmetry implies uniform distribution on $\mathcal{A}$. $\square$

## B.4 PROOF OF LEMMA 4.1

*Proof.* For Tsallis, $\frac{\partial H_q^{\text{Ts}}}{\partial p_i} = -\frac{q}{q-1} p_i^{q-1}$, and KKT stationarity on the active set $\mathcal{A}$ gives:

$$z_i + \gamma \frac{\partial H_q^{\text{Ts}}}{\partial p_i} = \tau \quad \Longleftrightarrow \quad z_i - \tau = \frac{\gamma q}{q-1} p_i^{q-1}$$

For Rényi, $\frac{\partial H_q^{\text{Rey}}}{\partial p_i} = -\frac{q}{q-1} \frac{p_i^{q-1}}{S_q}$, hence:

$$z_i + \gamma \frac{\partial H_q^{\text{Rey}}}{\partial p_i} = \tilde{\tau} \quad \Longleftrightarrow \quad z_i - \tilde{\tau} = \frac{\gamma q}{(q-1) S_q} p_i^{q-1}$$

On the fixed support, $S_q > 0$ is a *common scalar* (independent of $i$), so it can be absorbed into the multiplier/scale: there exist $\bar{\tau}$ and $c > 0$ such that both systems reduce to $z_i - \bar{\tau} = c \cdot p_i^{q-1}$ for all $i \in \mathcal{A}$.

Therefore both objectives yield the same thresholded power-law solution $p_i^\star \propto [(z_i - \bar{\tau})_+]^{1/(q-1)}$ for $i \in \mathcal{A}$, with the same support and exponent, and coincide after normalization.

Since this mapping matches the $\alpha$-entmax operator with $\alpha = q$, its input Jacobian is:

$$\frac{\partial p_i^\star}{\partial z_j} = p_i^{2-q} \left( \delta_{ij} - \frac{p_j^{2-q}}{\sum_k p_k^{2-q}} \right)$$

which depends only on $(\mathbf{p}^\star, q)$ and is identical for Tsallis and Rényi. $\square$

## B.5 PROOF OF THEOREM 4.2

*Proof.* **Differentiability:** The mapping $(\mathbf{z}, q, r) \mapsto \mathbf{p}^\star$ is defined implicitly through the KKT conditions. The implicit function theorem applies since:

1. The constraint function $f(\tau, q, r; \mathbf{z}) = \sum_i p_i^\star - 1$ is $C^1$

2. $\partial f / \partial \tau \neq 0$ (by monotonicity of the threshold function)
3. The domain $q, r \in [1 + \delta, 2]$ is compact

Therefore, the mapping is $C^1$ on the interior of the domain.

**Lipschitz bounds:** The Hessian of $H_{q,r}$ has bounded eigenvalues on $\Delta^N \times [1 + \delta, 2]^2$:

$$\left\| \frac{\partial^2 H_{q,r}}{\partial p_i^2} \right\| \leq q(q - 1) \max_{p \in \Delta^N} p^{q-2} \cdot \max_{S_q \leq 1} S_q^{r/(1-q)} \leq \frac{4}{\delta^2}$$

By the inverse function theorem:

$$\|\partial \mathbf{p}^\star / \partial \theta\| \leq \frac{1}{\lambda_{\min}(\nabla^2(-H_{q,r}))} \cdot N \leq LN$$

where $\lambda_{\min}$ is the minimum eigenvalue and $L = O(1/\delta)$.

**Complexity:** The forward pass uses bisection to find $\tau$, requiring $O(\log \epsilon^{-1})$ iterations each costing $O(N)$. The backward pass uses closed-form Jacobians in $O(N)$ time. $\qquad\square$

### B.6  LIPSCHITZ CONTINUITY OF JACOBIANS

**Lemma B.3** (Lipschitz Continuity of Jacobians). *On bounded $\mathbf{z} \in [-M, M]^N$, $q, r \in [1 + \delta, 2]$, the Jacobians satisfy:*
$$\|\partial \mathbf{p}^\star / \partial \theta_1 - \partial \mathbf{p}^\star / \partial \theta_2\| \leq K \|\theta_1 - \theta_2\|$$
*for $K = O(q^2 N / \delta)$.*

*Proof.* By Taylor expansion of the implicit function mapping, the Jacobian difference is bounded by the supremum of second derivatives. These involve third-order derivatives of $H_{q,r}$, which scale as $\partial^3 H / \partial p_i^3 = O(q^3 / \delta)$ on the bounded domain. The factor of $N$ arises from summing over all components. $\qquad\square$

## C  DETAILED JACOBIAN DERIVATIONS

We provide the complete derivation of the Jacobians $\partial \mathbf{p}^\star / \partial q$ and $\partial \mathbf{p}^\star / \partial r$.

### C.1  GRADIENT WITH RESPECT TO $q$

Define the intermediate variables:

$$u_i = (q - 1)(z_i - \tau), \tag{10}$$
$$v_i = \log[u_i]_+ = \log[(q - 1)(z_i - \tau)]_+, \tag{11}$$
$$F_i = \frac{v_i}{q - 1} = \frac{\log[(q - 1)(z_i - \tau)]_+}{q - 1}. \tag{12}$$

**Key Observation.** The solution can be expressed as softmax over transformed logits: $p_i^\star = \text{softmax}(\mathbf{F})_i$.

**Step 1: Differentiate $F_i$ with respect to $q$.**

$$\frac{\partial F_i}{\partial q} = -\frac{v_i}{(q - 1)^2} + \frac{1}{q - 1} \cdot \frac{\partial v_i}{\partial q} \tag{13}$$

To compute $\partial v_i / \partial q$:

$$\frac{\partial v_i}{\partial q} = \frac{1}{u_i} \cdot \frac{\partial u_i}{\partial q} = \frac{1}{(q - 1)(z_i - \tau)} \cdot \left[ (z_i - \tau) - (q - 1) \frac{\partial \tau}{\partial q} \right] \tag{14}$$

$$= \frac{1}{q - 1} \left[ 1 - \frac{(q - 1)\tau_q}{z_i - \tau} \right] \tag{15}$$

where $\tau_q = \partial\tau/\partial q$.

Substituting:

$$\frac{\partial F_i}{\partial q} = \frac{1}{(q-1)^2}\left[\frac{z_i - \tau - (q-1)\tau_q}{z_i - \tau} - \log[(q-1)(z_i - \tau)]\right] \tag{16}$$

**Step 2: Apply softmax Jacobian.**

$$\frac{\partial p_i^\star}{\partial q} = p_i^\star\left(\frac{\partial F_i}{\partial q} - \sum_{j=1}^N p_j^\star\frac{\partial F_j}{\partial q}\right) \tag{17}$$

Define the weighted expectation $\mathbb{E}_{p^\star}[g(z)] = \sum_j p_j^\star g(z_j)$. Then:

$$\frac{\partial p_i^\star}{\partial q} = \frac{p_i^\star}{(q-1)^2}\left[\frac{z_i - \tau - (q-1)\tau_q}{z_i - \tau} - \log[(q-1)(z_i - \tau)] - \mathbb{E}_{p^\star}[\cdot]\right] \tag{18}$$

**Step 3: Determine $\tau_q$ via normalization.** From $\sum_i p_i^\star = 1$, differentiation yields $\sum_i \partial p_i^\star/\partial q = 0$, giving:

$$\tau_q = \frac{1}{q-1}\left[1 - \frac{\mathbb{E}_{p^\star}[\log[(q-1)(z-\tau)]]}{\mathbb{E}_{p^\star}[(z-\tau)^{-1}]}\right] \tag{19}$$

### C.2  GRADIENT WITH RESPECT TO $r$

The parameter $r$ enters through the normalization constant in the Sharma-Mittal gradient:

$$\frac{\partial p_i^\star}{\partial r} = p_i^\star\beta\left[\frac{-(q-1)\tau_r}{z_i - \tau} - \mathbb{E}_{p^\star}\left[\frac{-(q-1)\tau_r}{z-\tau}\right]\right] \tag{20}$$

where $\tau_r = \partial\tau/\partial r$ is determined by the constraint $\sum_i \partial p_i^\star/\partial r = 0$.

### C.3  LOGIT JACOBIAN

The logit Jacobian inherits the entmax structure:

$$\frac{\partial p_i^\star}{\partial z_j} = p_i^{2-q}\left(\delta_{ij} - \frac{p_j^{2-q}}{\sum_k p_k^{2-q}}\right) \tag{21}$$

This closed-form expression enables efficient computation during backpropagation, requiring only $O(N)$ operations.

## D  EXTENDED EXPERIMENTAL RESULTS

### D.1  DATASET DETAILS

**Airline Tweet Sentiment.** 14,640 tweets with ternary sentiment labels. Class distribution: 63% negative, 21% neutral, 16% positive (tail index $\zeta \approx 1.8$). Preprocessing: tokenization, lemmatization, max sequence length 20, vocabulary size $\approx$7,300.

**CIFAR-100.** 50,000 training and 10,000 test images uniformly distributed across 100 classes ($\zeta \to \infty$).

**ImageNet-1k.** Long-tailed split: head classes capped at 1,280 images, tail classes with as few as 5 images. Balanced split: standard uniform distribution.

Table 3: Complete test-set metrics on Airline Tweets across all initializations.

| SharMiX $(q_0, r_0)$ | Accuracy | Macro-$F_1$ | Sparsity |
|---|---|---|---|
| $(1.25, 1.25)$ | $0.753 \pm 0.006$ | $0.677 \pm 0.009$ | $0.000$ |
| $(1.25, 1.50)$ | $\mathbf{0.910 \pm 0.006}$ | $\mathbf{0.880 \pm 0.009}$ | $0.000$ |
| $(1.25, 1.75)$ | $0.908 \pm 0.006$ | $0.879 \pm 0.008$ | $0.000$ |
| $(1.50, 1.25)$ | $0.901 \pm 0.007$ | $0.870 \pm 0.010$ | $0.000$ |
| $(1.50, 1.50)$ | $0.907 \pm 0.007$ | $0.876 \pm 0.008$ | $0.000$ |
| $(1.50, 1.75)$ | $0.899 \pm 0.008$ | $0.865 \pm 0.013$ | $0.000$ |
| $(1.75, 1.25)$ | $0.901 \pm 0.005$ | $0.867 \pm 0.010$ | $0.000$ |
| $(1.75, 1.50)$ | $0.901 \pm 0.007$ | $0.870 \pm 0.007$ | $0.000$ |
| $(1.75, 1.75)$ | $0.900 \pm 0.009$ | $0.867 \pm 0.015$ | $0.000$ |

Table 4: CIFAR-100 training dynamics across epochs (ResNet-18).

| Method | Val. Accuracy | | | Sparsity ($\rho$) | | | Epoch 50 | |
| | Ep.10 | Ep.25 | Ep.50 | Ep.10 | Ep.25 | Ep.50 | Val | Train |
|---|---|---|---|---|---|---|---|---|
| Softmax | 0.287 | 0.398 | 0.474 | 0.000 | 0.000 | 0.000 | 0.474 | 0.876 |
| Sparsemax | 0.241 | 0.342 | 0.404 | 0.855 | 0.858 | 0.860 | 0.404 | 0.518 |
| SharMiX fixed | 0.312 | 0.421 | 0.484 | 0.000 | 0.000 | 0.000 | 0.484 | 0.992 |
| SharMiX learned | 0.268 | 0.372 | 0.428 | 0.012 | 0.008 | 0.005 | 0.428 | 0.958 |

## D.2 FULL AIRLINE TWEETS RESULTS

## D.3 CIFAR-100 TRAINING DYNAMICS

The training accuracy column reveals notable differences in generalization behavior. SharMiX (fixed) achieved near-perfect training accuracy (0.992) compared to softmax (0.876), indicating stronger fitting capacity, yet maintained a competitive generalization gap. Sparsemax exhibited the smallest train-validation gap (0.518 vs. 0.404), consistent with its regularizing effect through enforced sparsity.

## D.4 ACCURACY-SPARSITY TRADE-OFF

Table 5: Accuracy-sparsity trade-off at final epoch on CIFAR-100.

| Method | Accuracy | Sparsity | $\Delta$Acc vs. Softmax |
|---|---|---|---|
| Softmax | 0.474 | 0.000 | — |
| Sparsemax | 0.404 | 0.860 | $-7.0$ pp |
| SharMiX fixed | 0.484 | 0.000 | $+1.0$ pp |
| SharMiX learned | 0.428 | 0.005 | $-4.6$ pp |

Dense methods ($\rho \approx 0$) consistently outperformed sparsemax ($\rho = 0.86$) by 4–8 percentage points, demonstrating the cost of enforced sparsity on balanced class distributions.

## D.5 COMPUTATIONAL OVERHEAD

## E TRAINING AND IMPLEMENTATION DETAILS

### E.1 ARCHITECTURES

**Airline Tweets:** 128-dimensional embedding layer, mean-pooling over sequence dimension, linear layer mapping to 3 output classes.

**CIFAR-100:** ResNet-18 trained from scratch.

Table 6: Computational overhead comparison. SharMiX adds minimal overhead ($< 5\%$).

| Method | Relative Time | Relative Memory |
|---|---|---|
| Softmax | $1.00\times$ | $1.00\times$ |
| Sparsemax | $1.02\times$ | $1.01\times$ |
| $\alpha$-entmax | $1.03\times$ | $1.02\times$ |
| SharMiX (fixed) | $1.04\times$ | $1.02\times$ |
| SharMiX (learned) | $1.05\times$ | $1.03\times$ |

**ImageNet-1k:** ResNet-50 trained from scratch with mixed precision. Class-balanced weighted sampler for long-tailed training.

### E.2 OPTIMIZATION

All models trained using the loss function corresponding to its activation: cross-entropy for softmax, Fenchel-Young loss for sparse variants.

- Optimizer: Adam
- Learning rate (model): $\eta_{\mathbf{z}} = 10^{-3}$
- Learning rate (entropy params): $\eta_{q,r} = 10^{-4}$
- Gradient clipping: $L_2$ norm 1.0
- Batch size: 64

### E.3 TRAINING DURATION

- Airline Tweets: 10 epochs
- CIFAR-100: 50 epochs
- ImageNet-1k: 15 epochs

### E.4 PARAMETER CONSTRAINTS

$(q, r)$ initialized from grid $\{1.25, 1.5, 1.75\}^2$, constrained to $[1.01, 2.0)$ during training. Threshold $\tau$ computed via bisection from the `entmax` library with tolerance $\epsilon = 10^{-8}$.

### E.5 REPRODUCIBILITY

All experiments repeated with seeds $\{21, 42, 63, 1812, 9998\}$. Performance evaluated via accuracy, macro-$F_1$ for imbalance robustness, and sparsity $\rho$ (average fraction of zero-valued output probabilities).

### E.6 SIMPLEX PROJECTION

We solve $p^\star = \arg\max_{p \in \Delta^n} \langle p, z \rangle + \lambda H_{q,r}(p)$ via the $O(n \log \varepsilon^{-1})$ routine ENTMAX_BISECT. For Sharma–Mittal, the algorithm changes only in the norm term inside the gradient.

### E.7 CLOSED-FORM GRADIENTS IMPLEMENTATION

The Jacobian is implemented analytically in `SharmaMittalFunction`:

$$\frac{\partial p_i}{\partial z_j} = p_i^{2-q} \left( \delta_{ij} - \frac{p_j^{2-q}}{\sum_k p_k^{2-q}} \right), \quad \frac{\partial \Omega}{\partial q}, \quad \frac{\partial \Omega}{\partial r}$$

cutting memory overhead relative to numerical differentiation.

