# OpenReview forum: "Learning When to Be Sparse: Adaptive Activations via Two-Parameter Entropy"
_ICLR.cc/2026/Workshop/Sci4DL — Sci4DL 2026_

### Official Review · Reviewer_NTo2 · 2026-02-25

**Fit:** 3
**Significance:** 2
**Confidence:** 2

**Summary:**

This paper introduces **SharMiX**, a novel, two-parameter learnable activation function designed to replace the traditional Softmax operator. Softmax, derived from Shannon entropy, strictly produces dense probability distributions, allocating non-zero probability mass even to highly improbable classes—a significant limitation in heavy-tailed or imbalanced datasets. While existing sparse alternatives like Sparsemax or $\alpha$-entmax (based on Tsallis entropy) can output exact zeros, their single parameter rigidly couples the distribution's curvature with its sparsity level.

To overcome this, the authors propose an elegant mathematical framework based on the two-parameter Sharma-Mittal entropy. SharMiX successfully unifies Softmax, Tsallis/$\alpha$-entmax, and Rényi entropies into a single differentiable mapping. The core technical contribution is the derivation of closed-form, Lipschitz-continuous Jacobians for both the input logits and the two entropy parameters ($q, r$). This allows the network to learn *when* to be sparse and *when* to be dense end-to-end via backpropagation. Empirical results on CIFAR-100, ImageNet-1k, and an imbalanced text classification task demonstrate that SharMiX automatically adapts its sparsity level to the underlying data distribution (e.g., highly sparse for long-tailed ImageNet, but dense for balanced CIFAR-100) with minimal computational overhead.

**Strengths:**

*   **Elegant and Solid Theoretical Foundation:** The theoretical formulation is deeply rooted in statistical physics (generalized entropies). Providing a unified mathematical framework that encompasses Softmax and its sparse variants (visualized beautifully in the $(q, r)$ parameter space in Figure 1) is a strong first-principles contribution that aligns perfectly with the goals of SciForDL.
*   **Impressive Algorithmic Tractability:** Deriving efficient, closed-form gradients for a complex two-parameter projection onto the probability simplex is a non-trivial mathematical achievement. The authors successfully translated a heavy theoretical concept into a practical neural network layer with negligible computational overhead (< 5% relative time/memory increase, as shown in Table 6).
*   **Compelling "Data-Driven Adaptivity":** The empirical validation (particularly Figure 2) provides striking evidence of the method's core claim. Watching the model automatically discover that it needs high sparsity ($\rho > 0.91$) for a long-tailed dataset, while collapsing back to a dense Softmax-like state ($\rho \to 0$) for a balanced dataset, is highly convincing.

**Suggestions:**

While the theoretical derivations are rigorous and the initial empirical results are promising, the paper would benefit significantly from clarifications regarding its scope, improved visual intuition, and expanded experimental settings. I strongly recommend addressing the following points:

**1. Clarifying the Scope: "Activations" vs. "Probability Mappings on the Simplex"**
In the abstract and introduction, the authors broadly state that traditional activations (referring to Softmax) are inherently dense. However, in the wider deep learning context, hidden-layer activation functions like ReLU are famously utilized precisely because they induce *activation sparsity* (by outputting exact zeros for negative inputs).
*   *Suggestion:* The authors should explicitly distinguish their work from general hidden-layer activations early in the Introduction. It must be clarified that SharMiX is specifically designed as a **probability mapping function (or normalizer)** constrained to the probability simplex ($\Delta^N$), typically used in output layers or attention mechanisms, rather than as a general replacement for ReLU-like hidden activations.

**2. Missing Visualizations of the SharMiX Function**
While Figure 1 elegantly visualizes the $(q, r)$ parameter space, the paper lacks a fundamental visual aid expected when introducing any new activation function: a plot of the function's input-output curve.
*   *Suggestion:* The authors must include a 2D plot (e.g., in Section 4 or the Appendix) showing the input logits $z$ on the x-axis and the resulting probability output $p^*$ on the y-axis, for various fixed values of $q$ and $r$. This visualization is absolutely crucial for readers to intuitively grasp how modifying $(q, r)$ simultaneously shifts the "hard-zero threshold" (controlling sparsity) and alters the "steepness/curvature" of the non-zero region. Without this, the geometric intuition behind the complex Sharma-Mittal formulation remains overly abstract.

**3. Enhancing the Intuitive Explanation of $q$ vs. $r$ Dynamics**
In Section 6 (Discussion), the authors state that "$r$ drives adaptation while $q$ remains stable." However, due to the abstract nature of the dual-parameter entropy, it is difficult for readers to intuitively understand *why* this occurs during optimization.
*   *Suggestion:* Provide a brief intuitive or visual explanation (perhaps an ablation tracking the gradients of $q$ and $r$ over time) to explain the distinct roles of these parameters during end-to-end training. Why does the gradient strongly push $r$ to control sparsity, while leaving $q$ to maintain the shape/curvature? This would greatly enhance the paper's educational value.

**4. Expanding Experiments to Natural Language Generation (Attention Mechanisms)**
The current experiments are limited to classification tasks. However, the most profound impact of sparse mapping functions (like Sparsemax or $\alpha$-entmax) has historically been in the **attention mechanisms** of Transformer models or Neural Machine Translation (NMT), where long-tailed distributions (Zipf's law) over vocabularies are extreme.
*   *Suggestion:* As the authors mention in their Conclusion, applying SharMiX to attention mechanisms is a natural next step. While a full LLM pretraining run is likely out of scope for a Workshop paper, demonstrating SharMiX's adaptive sparsity within the attention layer of a smaller-scale Transformer (e.g., on a standard NMT benchmark or language modeling task) would dramatically increase the impact and relevance of this work for the broader community.

---

### Official Review · Reviewer_y8ZS · 2026-02-27

**Fit:** 2
**Significance:** 2
**Confidence:** 2

**Summary:**

This paper proposes SharMiX, a two-parameter probability mapping derived from Sharma–Mittal entropy. The motivation is that softmax can be viewed as the maximizer of a Shannon-entropy-regularized objective, which always yields dense outputs, while existing sparse alternatives (such as sparsemax and α-entmax) use a single parameter that may couple sparsity with the shape/curvature of the mapping. SharMiX introduces two learnable entropy parameters $(q,r)$ and formulates prediction as an entropy-regularized maximization over the simplex. The solution is presented in a thresholded power-law form with a normalization threshold $\tau$, where $q$ controls sparsity through $β=1/(q−1)$. To enable end-to-end learning of $(q,r)$, the paper derives closed-form Jacobians using implicit differentiation of the normalization constraint, and states differentiability, Lipschitz Jacobian bounds, and computational complexity guarantees. Empirically, SharMiX is evaluated on an imbalanced text dataset (Airline Tweets), a balanced image dataset (CIFAR-100), and ImageNet-1k under both long-tailed and balanced regimes, showing high sparsity only in the long-tailed ImageNet setting while collapsing to dense outputs on balanced regimes.

**Strengths:**

The paper addresses a clear and practical question: how to adapt the output mapping between dense (softmax-like) and sparse (entmax-like) behavior depending on the data regime. The use of Sharma–Mittal entropy is conceptually appealing because it unifies Shannon, Rényi, and Tsallis families in a two-parameter space, and the paper provides explicit definitions and limiting-case connections. The methodological contribution of deriving analytic gradients with respect to both logits and entropy parameters is useful for making such layers trainable, and the paper discusses computational aspects (bisection for $\tau$, closed-form Jacobians, and reported small overhead). Empirically, the ImageNet long-tailed vs balanced comparison provides an intuitive demonstration that the learned behavior can differ across regimes.

**Suggestions:**

My major concern is that, in the main reported results, SharMiX often converges to fully dense outputs on Airline Tweets and CIFAR-100 (sparsity =0), while the learned-parameter variant underperforms softmax on CIFAR-100 (0.428 vs 0.474) with only negligible sparsity. This makes the claimed “accuracy–sparsity trade-off navigation” less convincing beyond the long-tailed ImageNet split. It would strengthen the paper to include stronger baselines and fair tuning: for example, tuned α-entmax (varying α), learned-α variants (as in prior work), and temperature scaling baselines, to separate gains from simple rescaling effects. The role of parameter $r$ in the forward mapping is also not fully clear from the main equations (the presented power-law form highlights $q$ through 𝛽); clarifying exactly how
𝑟 influences the solution (not only its gradients) would improve readability and trust in the mechanism. Finally, the paper itself notes limited coverage of distributions and sensitivity to initialization; more systematic sweeps over initialization, learning rates for $(q,r)$, and reporting of stability across seeds on all benchmarks would help.

---

### Meta-Review · Area_Chair_EhB4 · 2026-03-01

**Recommendation:** Accept

**Metareview:**

The paper is a good fit for the workshop, addresses an important research question, and makes important contributions. I recommend acceptance.

---

### Decision · Program_Chairs · 2026-03-02

Accept